



# Quasi-steady circulation regimes in the Baltic Sea

**Taavi Liblik[1], Germo Väli[1], Kai Salm[1], Jaan Laanemets[1], Madis-Jaak Lilover[1], Urmas Lips[1]**
[1]Department of Marine Systems, Tallinn University of Technology, Tallinn, Estonia
**\* Correspondence:**
Taavi Liblik
taavi.liblik@taltech.ee
**Keywords: Circulation, ADCP, underwater glider, Baltic Sea, boundary current, geostrophic
current, upwelling-downwelling.**
**Abstract.** Circulation plays an essential role in the creation of physical and biogeochemical fluxes in
the Baltic Sea. The main aim of the work was to study the quasi-steady circulation patterns under
prevailing forcing conditions.
Six months of continuous vertical profiling and fixed-point measurements of currents, two monthly
underwater glider surveys, and numerical modelling were applied in the central Baltic Sea. The vertical
structure of currents was strongly linked to the location of the two pycnoclines: the seasonal
thermocline and the halocline. The vertical movements of pycnoclines and velocity shear maxima were
synchronous. The quasi-steady circulation patterns were in geostrophic balance and high-persistent.
The persistent patterns included circulation features such as upwelling, downwelling, boundary
current, and sub-halocline gravity current. The patterns had a prevailing zonal scale of 5–60 km and
considerably higher magnitude and different direction than the long-term mean circulation pattern.
Northward (southward) geostrophic boundary current in the upper layer was observed along the eastern
coast of the central Baltic in the case of southwesterly (northerly) wind. The geostrophic current at the
boundary was often a consequence of wind-driven, across-shore advection.
The sub-halocline quasi-permanent gravity current with a width of 10–30 km from the Gotland Deep
to the north over the narrow sill separating the Farö Deep and Northern Deep was detected in the
simulation, and it was confirmed by an Argo float trajectory. According to the simulation, a strong
flow, mostly to the north, with a zonal scale of 5 km occurred at the sill. This current is an important
deeper limb of the overturning circulation of the Baltic Sea. The current is stronger with northerly
winds and restricted by the southwesterly winds.
The circulation regime has an annual cycle due to seasonality in the forcing. Boundary currents are
stronger and more frequently northward during the winter period. The sub-halocline current towards
the north is strongest in March–May and weakest in November–December.



# 1    Introduction

Current structure is an important player in the physical and biogeochemical fluxes in ocean. The semi-enclosed, shallow, brackish Baltic Sea has a strong but variable vertical stratification characterized by two pycnoclines: the permanent halocline and the seasonal thermocline (Leppäranta & Myrberg, 2009). Three-layer structure occurs in summer and consists of warm and fresh upper mixed layer, cold and saltier intermediate layer, and warmer and saltiest deep layer. Water column is mixed up to the permanent halocline at 60–80 m depth and cold intermediate water forms during winters. Stratification through the two pycnoclines impedes vertical mixing, and transport of substances between the layers is limited. The role of tides is marginal in the Baltic Sea. Lateral flows play an important role in distributing the water properties.

Water-mass circulation of the Baltic Sea is determined by the saline water inflow from the North Sea and freshwater input from the catchment area. The interaction of the fresher and saltier waters forms the so-called Baltic haline conveyor belt (Döös et al., 2004). The belt consists of saltier water transport and signal propagation in the deep layer towards the north-eastern end of the Baltic (Liblik et al., 2018; Väli et al., 2013); upward salt flux through vertical mixing and transport (Reissmann et al., 2009), and outflow of the mix of riverine and saltier water in the upper layer (Jakobsen et al., 2010). The conveyor determines salinity, stratification and other important characteristics for the pelagic ecosystem.

The largest basin in the sea, the Baltic Proper (Fig. 1a) is a source for the deep waters of the Gulf of Riga, Gulf of Finland and Gulf of Bothnia. Permanent oxygen depletion has expanded in recent decades in the Baltic Sea, forming one of the largest dead zones in the global ocean (e.g. Carstensen et al., 2014). Only Major Baltic Inflows (Matthäus & Franck, 1992; Mohrholz, 2018) ventilate the deep layers of the southern and central Baltic Proper (Holtermann et al., 2017) but increase hypoxia in the Northern Baltic Proper and Gulf of Finland (Liblik et al., 2018).

The basin-scale pattern of the long-term mean circulation in the Baltic Proper is cyclonic as demonstrated by several modelling studies (Hinrichsen et al., 2018; Jedrasik et al., 2008; Jędrasik & Kowalewski, 2019; Meier, 2007; Placke et al., 2018). The mean circulation is to the north along the eastern coast of the Baltic Proper and to the south along the eastern and western coast of Gotland Island (Meier, 2007; Placke et al., 2018). The turning area for this basin-wide cyclonic circulation cell in the north is between 59 to 59.5° N (Meier, 2007). The zonal center of the cyclonic flow in the Eastern Gotland Basin is in the Gotland Deep (Placke et al., 2018). The cyclonic structure exists from the bottom to the surface (Placke et al., 2018), although lateral structure and magnitude of the flow vary among different models (Placke et al., 2018). It is important to note that all forementioned descriptors of the long-term mean flow rely on numerical simulations and lack support from observations. However, a consistent northward low-frequency current along the eastern slope of the Gotland Deep at 204 m depth has been reported (Hagen & Feistel, 2004). Placke et al. (2018) compared simulated currents with these measurements. All model simulations showed the mean meridional northward current velocity in the range of 0–1 cm s$^{-1}$ (actually, three models out of four had values of 0.0–0.1 cm s$^{-1}$) while the measurements gave the mean northward velocity of 3 cm s$^{-1}$ (Hagen & Feistel, 2004). Thus, the long-term mean flow to north in the deep layer was much stronger than the simulated mean current.

Temporal variability of currents in the Baltic Sea is very high as a reaction to atmospheric forcing. Near-shore Eulerian current observations (Sokolov & Chubarenko, 2012) and drifter experiments



(Golenko et al., 2017; Krayushkin et al., 2019) conducted in the southern Baltic Proper showed a strong
correlation between wind and surface currents. Current velocity spectra in the Baltic include seiches
and tides with different periods from 11 h to 31 h and inertial motions with a period of about 14 h
(Jönsson et al., 2008; Lilover et al., 2011; Suhhova et al., 2018).
The vertical current structure through thermocline and halocline has not been rigorously studied by the
in-situ observations in the Baltic Proper. Moreover, despite a considerable effort to reveal the spatial,
long-term mean circulation patterns based on the simulations, not much has been done to study
temporal developments of currents in the synoptic (mesoscale) and seasonal timescales in the Baltic
Proper. In the present work, we address this shortage of knowledge.
Permanent circulation systems, such as boundary currents or subtropical gyres, are key processes that
determine transport in the open ocean (e.g. Macdonald, 1998). Although there are no permanent
currents in the Baltic Sea, we hypothesize that under stable wind forcing and stratification conditions,
a steady circulation regime prevails in the time-scale of days to weeks and has a much greater
magnitude than the mean current structures. These quasi-steady circulation features could be related to
the downwelling and upwelling processes or appear as a boundary current or a gravity current under
the halocline.
Following a description of the methods used, we present an analysis of (1) boundary current under
variable wind forcing and stratification, (2) quasi-permanent circulation patterns, and (3) sub-halocline
current. The analysis of observational and simulation results is followed by discussion and conclusions.

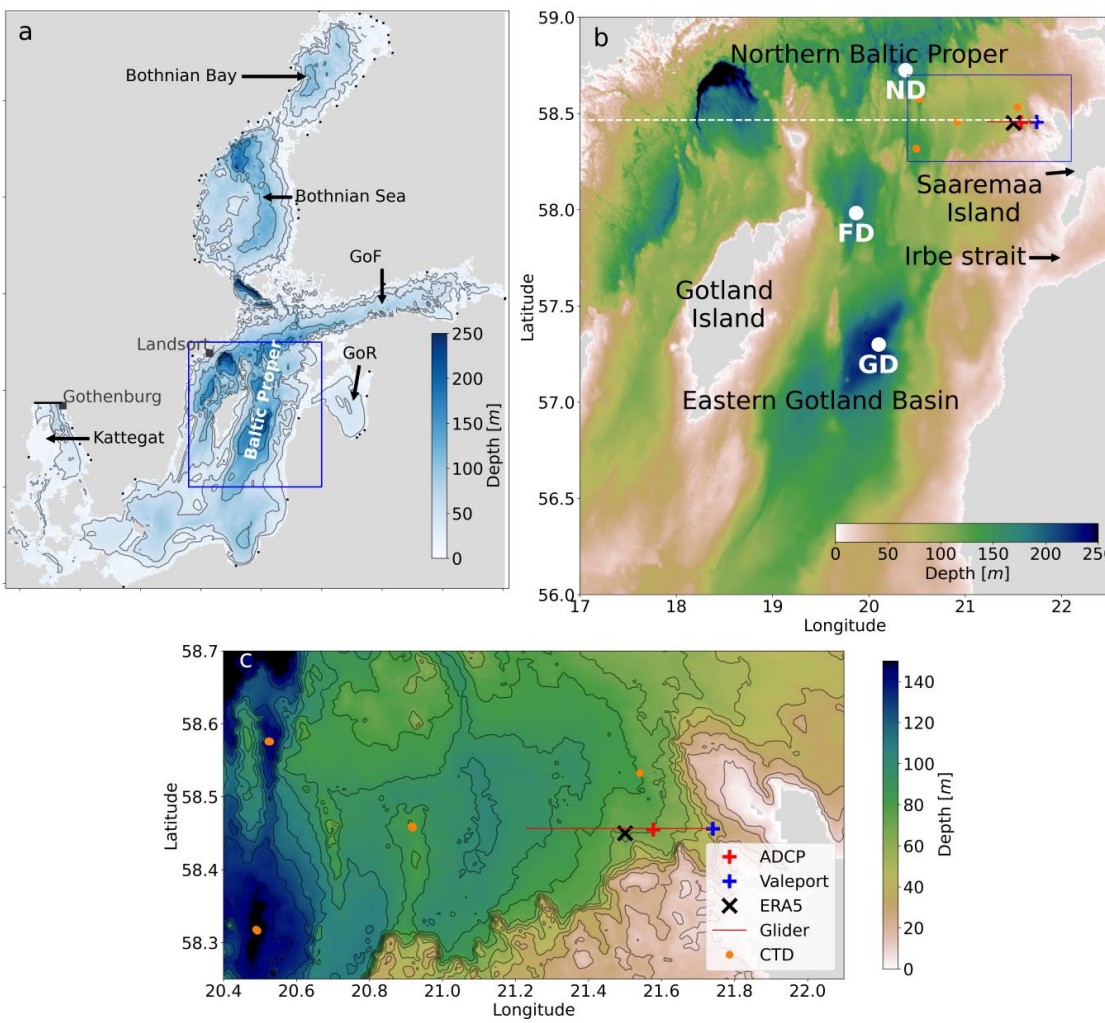


**Figure 1.** (a) Map of the Baltic sea and model domain. Shown are the locations of the open boundary of the
model domain in the Kattegat (bold black line), Landsort and Gothenburg sea level stations, Baltic Sea rivers
used in the model (black dots) and study area (black box). (b) Close-up of the study area. Locations of ADCP
and Valeport moorings, CTD measurements, glider section, the center of the cell of ERA5 wind data, and zonal
section along the latitude of the ADCP location in the Nortern Baltic Proper (white dashed line) are presented.
Gotland Deep (GD), Fårö Deep (FD) and Northern Deep (ND) are also shown. (c) Close view of the moorings
and CTD measurement locations, glider section, and local topography are shown.


**2   Data and methods**
**2.1 Observations and data products**
A bottom mounted current profiler ADCP 300 kHz (Teledyne RDI) and model 106 current meter
(Valeport Ltd) (hereinafter referred to as Valeport) were deployed at the end of February to the west of
Saaremaa Island (Fig. 1b and c). Valeport was mounted at 5 m depth, while the sea bottom depth in its



location was 41 m. The sea depth in the ADCP location was 71 m and velocities were measured with
vertical depth interval of 2 m in the depth range of 10–68 m. Current velocity profiles were recorded
as average of 1 h. The quality of the current velocity data was checked following the procedure
developed by Book (et al., 2007). Valeport recorded current velocity with 10 min intervals. A Seabird
SBE 16*Plus* V2 CTD SEACAT conductivity and temperature recorder was deployed together with the
ADCP, but it hung 4 m above the sea bottom, i.e., at a depth of 67 m. SBE 16*Plus* sensors were
calibrated by the manufacturer before the deployment.
Repeated CTD profiles onboard R/V Salme were collected using an OS320 CTD probe (Idronaut S.r.l.)
in the Northern Baltic Proper (see Fig. 1b and c) from 30 January to 4 August 2020.
Argo float deployment was arranged by the Finnish Meteorological Institute (Siiriä et al., 2019) from
15 August 2013 to 15 August 2014 and the trajectory data was derived from the Argo-based deep
displacement dataset (Ollitrault & Rannou, 2013). The dataset was downloaded on 15 March 2021 at
https://www.seanoe.org/data/00360/47077/.
In 2020, two glider missions were conducted in the Northern Baltic Proper. The Slocum G2 Glider
collected oceanographic data along the E–W oriented 27 km long section (Fig. 1b and c). The
easternmost point of the glider track was approximately 7 km off the shoreline and the section was
located at the sloping bottom where sea depth gradually deepened westward from 40 m to 90 m. The
first mission was carried out from 28 February to 22 March 2020 and the second one from 4 August to
2 September 2020. Both ascending and descending profiles were recorded and altogether over 8000
profiles were gathered. The glider moved at a horizontal speed of $0.33\pm0.08$ m s$^{-1}$. On average, a profile
took $8.0\pm0.9$ min to complete 80–90 m deep profile and the average distance between the profiles near
the surface was $301\pm46$ m. Both the sampling time and the distance were decreased by half in the
shallow part of the section.
Preliminary glider data processing included the standard quality control (impossible date and location
test, range tests for the sensors) and accounting for the response time of the sensors and the thermal
lag. First, a linear time shift was applied to temperature and conductivity considering the misalignment
with pressure. Temperature was re-aligned by 1.4 s and conductivity by 0.9 s for the mission conducted
in the spring and respectively by 1.6 s and 1.1 s for the mission in the summer. The parameters were
chosen by comparing consecutive profiles focusing on the depth range around the greatest gradient. It
was assumed that successive profiles correspond to the same water mass. We followed Mensah et al.
(2009) to remove the thermal lag effect and found optimal coefficients for the temperature error
amplitude, $\alpha$, and time constant, $t_c$, by comparing consecutive TS-profiles. The satisfying results were
obtained in the case of $\alpha = 0.0025$ and $t_c = 10$ s for the earlier mission and $\alpha = 0.055$ and $t_c = 12$ s for
the following one. The profiles were averaged on a 0.5 dbar vertical grid after processing the raw data.

Sea surface temperature was derived from the Copernicus Marine Service product
SST_BAL_SST_L4_REP_OBSERVATIONS_010_016 with a horizontal resolution of 0.02 x 0.02
degrees. Mean difference between the product and in-situ data sources has been in the range of –0.12
to –0.21 °C and root mean square error from 0.43 to 0.88 °C depending on the data sources according
to           the           quality           information           document
(https://catalogue.marine.copernicus.eu/documents/QUID/CMEMS-SST-QUID-010-016.pdf,
accessed 19 August 2021).



Hourly, 10 m level wind velocities of ERA5 reanalysis data (Hersbach et al., 2020) at the cell with the
size 0.25°x0.25° from 1979 to 2020 (see Fig. 1 for location) were used in the analyses.

## 2.2 Modelling

Numerical model GETM (General Estuarine Transport Model, Burchard & Bolding, 2002) has been
applied to simulate the circulation and temperature/salinity distribution in the northeastern Baltic Sea.
GETM is a primitive equation, three-dimensional model with free surface and $k$–$\varepsilon$ turbulence model
for vertical mixing by coupling the hydrodynamic part with GOTM (General Ocean Turbulence Model,
Umlauf & Burchard, 2005).
Model domain covered the whole Baltic Sea with the open boundary situated in the Kattegat region
(Fig. 1a). The horizontal grid spacing of the model was 0.5 nautical miles (926 m) and 60 vertically
adaptive coordinates (Hofmeister et al., 2010; Gräwe et al. 2015) were used. Sea surface height from
Gothenburg station has been used as the boundary condition to control the barotropic in- and outflow
from the Baltic Sea, while the temperature and salinity were nudged towards monthly climatological
profiles (Janssen et al., 1999) along the open boundary.
Data from the Estonian version of the operational model HIRLAM (High Resolution Limited Area
Model) maintained by the Estonian Weather Service and giving forecasts with hourly resolution
(Männik and Merilain, 2007) were used to calculate the momentum and heat flux at the sea surface.
Climatological runoff of the Baltic Sea rivers with inter-annual variability added from the values
reported to the HELCOM (Johansson, 2016) was used. Simulation covered period from April 2010 to
September 2020, and initial temperature and salinity fields were taken from the CMEMS (Copernicus
Marine Service) re-analysis product for the Baltic Sea.
The same setup of the model was previously used in Zhurbas et al., (2018) and Liblik et al. (2020) and
more details about the model setup are given there. Zhurbas et al. (2018) validated the salinity and
temperature values in the central Baltic Sea along with the sea surface height at Landsort station and
compared the near-bottom current statistics with the long-term observations in the Gotland Deep.
Liblik et al. (2020) validated the simulated wintertime sea surface temperature and salinity in the Gulf
of Finland and compared the observed mixed layer depth with the simulations. In this study, we will
present the comparison of simulated and observed currents in the Northern Baltic Proper.

## 2.3 Calculations

Isohaline 9 g kg$^{-1}$ was selected to define the center of the halocline (CH) depth since the halocline was
steepest around this salinity value according to the salinity profiles. To estimate the center of halocline
depth based on single level salinity time-series measured by the SBE 16*Plus*, and twelve CTD profiles
collected by the RV Salme in the Northern Baltic Proper (see Fig. 1b) from 30 January to 4 August
2020 were used. Salinity profiles were vertically normalized by subtracting the depth of the CH at each
profile. Next, the mean salinity profile in the normalized depth coordinates was calculated (Fig. 2). The
mean normalized depth and salinity relationship were used to derive the CH depth from the SBE 16*Plus*
salinity time-series at 67 m depth. If salinity was lower (higher) than 9 g kg$^{-1}$, the CH was deeper
(shallower) than 67 m according to the mean depth-salinity curve (Fig. 2). Maximum depth of the
neighboring sea area, 88 m, was defined as the maximum depth of the CH.



In this study the *x*-axis is positive eastward, the *y*-axis is positive northward, and the *z*-axis is positive
upward (*z*=0 at the sea surface), *u* and *v* are horizontal velocity components.
The baroclinic components of the geostrophic velocity ($u_g$ *and* $v_g$) can be deduced from the
hydrographic data. Considering the dynamic method, the geostrophic relationships are as follows
$v_g = \frac{1}{f}\frac{\partial \Phi}{\partial x}$
$u_g = -\frac{1}{f}\frac{\partial \Phi}{\partial y}$
The geopotential, $\Phi$, is proportional to the dynamic height, D, as
$\Phi = gD$
where *g* is the gravitational acceleration and *f* is the Coriolis parameter.
The dynamic height can be determined from the temperature and salinity (density) profiles.
The relative geostrophic velocity was evaluated using dynamic height anomaly relative to a reference
pressure (McDougall & Barker, 2011). The geopotential slope of an isobaric surface expresses the
horizontal pressure gradient. A zonal glider track enabled to calculate the meridional velocity profile
of the geostrophic flow. The meridional geostrophic velocity was calculated also from the GETM
simulation data. The reference level was set at 70 dbar. The shallower profiles were included using the
stepped no-motion level method described in Rubio et al. (2009). Since velocity is not zero at the 70
dbar level, the calculated geostrophic velocities $V_{GEO-DENS-glider}$ and $V_{GEO-DENS-GETM}$ described in
subchapter 3.1 represent relative velocities to the no-motion 70 dbar level. Both variables represent an
averaged velocity at an extent of 10 km zonal scale around ADCP position.
To compare the simulated geostrophic velocity profiles with the measured ADCP velocity profiles, the
relative geostrophic velocity at the sea surface (calculated relative to 70 dbar using simulated density
profiles) was aligned with the geostrophic velocity due to the sea level gradient from the model
simulation ($V_{GEO-SL-GETM}$). Sea level gradient was estimated from linear regression fit of sea level
anomalies at a horizontal scale of 10 km. The difference (vector) between the density-estimated and
the sea level estimated geostrophic velocity at the sea surface was applied to the whole geostrophic
velocity profile under the assumption that the geostrophic current at the surface is determined by the
differences in the sea level exclusively. Adjusted geostrophic velocity profiles were presented as $V_{GEO-ADJ-GETM}$ in subchapter 3.2.

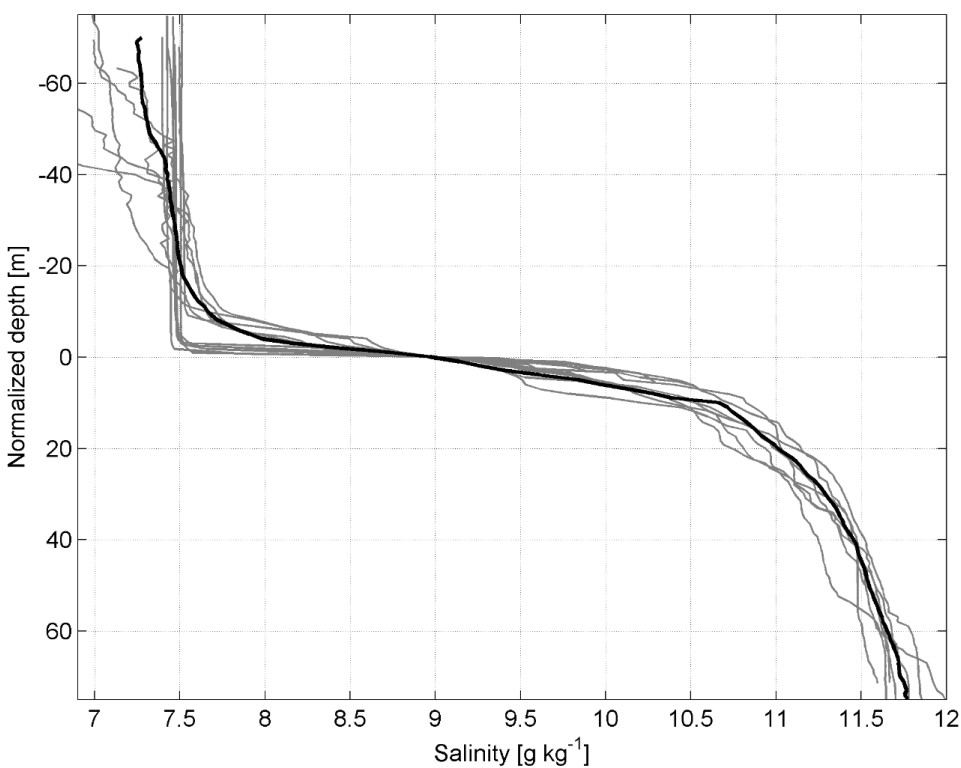


**Figure 2.** Vertically normalized salinity profiles from 30 January to 4 August 2020 in the Northern Baltic Proper (see Fig. 1b). Bold black line represents the mean salinity profile.

The direct influence of wind forcing on the subsurface currents was ascertained using the classical Ekman model based on the balance of the frictional and Coriolis forces (Ekman, 1905). Wind stress vector $\boldsymbol{\tau}$ as the Ekman model input parameter was calculated using ERA5 (Fig. 1b and c) wind data: $\boldsymbol{\tau} = \rho_{air}cd|\mathbf{U}|\mathbf{U}$, which were prior low-pass filtered with cut-off 36 hours to exclude periodic processes. Here $\mathbf{U}$ is the wind velocity vector at 10 m height, cd is the drag coefficient and was parameterized as proposed by (Wu, 1980): cd=$(0.8+0.065|\mathbf{U}|)\times10^{-3}$, $|\mathbf{U}|$ is the wind velocity vector module and $\rho_{air}$ is the density of air. The eddy viscosity used in the model was calculated according to (Csanady, 1981): $\nu = |\boldsymbol{\tau}|/200f$, where $|\boldsymbol{\tau}|$ is the wind stress vector module. The model outputs are the vertical profiles of wind-induced current velocity components.

The temporal development in the vertical current structure is presented as the time-series of vertical current shear squared $s^2 = (\partial u/\partial z)^2 + (\partial v/\partial z)^2$.

Persistency of the current is defined as the ratio between vector and scalar current speeds:

$$R = \frac{\sqrt{u^2+v^2}}{\frac{1}{N}\Sigma\sqrt{u_n^2+v_n^2}}.$$






Current and wind velocity components are presented as 36-h and 10-day low-passed time-series. The
fourth-order Butterworth filter was used for low-pass filtering.

## 3 Results

### 3.1 Boundary current under variable wind forcing

Statistics of the 6 months (1 March–1 September 2020) ADCP deployment revealed the persistency of
currents between 32 and 42%, with the highest persistency in the 20–40 m depth range (Table 1). Mean
and maximum hourly measured speeds were higher in the uppermost bin at 11 m depth, 11 and 48 cm
s$^{-1}$, respectively and lower in the near-bottom layer, 7 and 34 cm s$^{-1}$. The mean $u$- and $v$-components
were positive in all depths showing the mean flow to the NE sector.
From the flow structure point of view the ADCP current velocity time series can be divided into two
periods: 1) from March until mid-April, when barotropic regime prevailed, 2) from mid-April until
September, when layered flow dominated (Fig. 3a and b). One can also see the coincidence of the
current $u$- and $v$-components in the uppermost and deepest bin during the first period (Fig. 3c and d)
except a short period at the end of March. Discrepancies between the two layers afterwards illustrated
the layered, baroclinic nature of the flow. The flow regime reacted well to wind forcing. Barotropic
flow to the northeast prevailed as a result of southwesterly winds until mid-April (Fig. 4). Only during
the last week of March, when wind was from northerly directions, a strong southerly current was
observed. Similar temporal patterns appeared in the upper layer in the stratified period. Alteration of
positive and negative meridional velocities was related to the prevailing wind direction. These
tendencies were evident both in the ADCP and Valeport locations. Deep layer current was directed to
the east, i.e., onshore, when southerly flow occurred in the upper layer and to the west or southwest,
when the current to the northeast prevailed. These are signs of the layered structure of the coastal
upwelling and downwelling.
The most frequent current direction in the upper layer (11 m depth) was 40° at the ADCP location. To
estimate the relationship between the low-frequency (10-day low-passed) current component and wind,
we calculated the correlation between the 40° current velocity component ($c_{40}$) in the upper layer and
wind speed from different directions with different time lags. The best correlation ($r^2$=0.65, p<10$^{-100}$,
n=4473) was found with the wind from the south, specifically towards 10° ($w_{10}$), applying a 3-day time
lag. This, on the one hand, corresponds to Ekman's theory, however, on the other hand, the 3-day delay
is rather long. Probably it can be explained by the mixed effect of wind on the surface currents. The
momentum flux created by wind impacts the current field fast. The correlation without delay is
relatively high ($r^2$=0.55, p<10$^{-100}$, n=4473) as well. The flow resulting from the sea level gradient and
due to the inclination of isopycnal surfaces are also a consequence of wind but develop slower.
Time series of $c_{40}$ reveal negative values from mid-April until the end of June (Fig. 3e). Before mid-
March and in July–August, the $c_{40}$ was mostly positive. The main course of $w_{10}$ and $c_{40}$ coincided well,
but discrepancies occurred in the details. For instance, negative $c_{40}$ occurred when $w_{10}$ was positive in
the ADCP location in the last third of March and first half of May. The mean values of $w_{10}$ and $c_{40}$
during the measurements were 0.6 m s$^{-1}$ and 3.2 cm s$^{-1}$, respectively. Considering the linear relation
between the two variables, the 1979–2020 mean $w_{10}$ = 1.1 m s$^{-1}$ corresponds to $c_{40}$ = 4.2 cm s$^{-1}$.
The most frequent current direction was 350° at the Valeport location. The discrepancy between the
dominant flow direction at the ADCP and Valeport locations is related to the topographic features (Fig.
1). However, from the wider Baltic Sea dynamics point of view the meridional current component is
important to investigate. To study the temporal developments of the meridional current, we next



analyze the measured and simulated meridional current components at 11 m depth at the ADCP
location, $V_{ADCP}$ and $V_{GETM}$. We also calculated the geostrophic component $V_{GEO-SL-GETM}$ of the current
velocity from the simulated sea level gradient, relative geostrophic meridional current component
($V_{GEO-DENS-GETM}$) at 11 m depth based on simulated temperature and salinity data in the section and
same for the glider temperature and salinity data ($V_{GEO-DENS-glider}$). We also calculated mean Ekman
current $u$- and $v$-components in the depth range 0–10 m $U_{Ekman}$ and $V_{Ekman}$, respectively. All parameters
are 36-h low-passed filtered.
Overall, the simulated $V_{GETM}$ reasonably well follows the temporal changes in measured $V_{ADCP}$ (Fig.
5). $V_{GETM}$ tends to have smaller values than $V_{ADCP}$, which means that the meridional component of
simulated velocity is biased southward. Sometimes, e.g., in June and August, the discrepancies are
considerable. Geostrophic current $V_{GEO-DENS-GETM}$ was very small, and $V_{GEO-DENS-glider}$ was practically
zero in March (Fig. 5b) as the water column was mixed down to the reference depth of the geostrophic
current calculation. Since the end of March, overall temporal developments in the meridional current
($V_{ADCP}$ and $V_{GETM}$) and its geostrophic components ($V_{GEO-DENS-GETM}$), ($V_{GEO-SL-GETM}$) and $V_{GEO-DENS-}$
$_{glider}$) in August match quite well (Fig. 5a and b). This can be related to the multiple effects of wind.
South-westerly wind resulted in the Ekman current towards the eastern coast of the Northern Baltic
Proper. This caused, first, a sea level gradient across the basin (higher near the coast), which induced
barotropic current to the north. Secondly, it evoked downwelling along the coast and resulted in a
vertical gradient of the geostrophic current. Such events were detected at the beginning of April and
July, when strong southwesterly winds blew (Fig. 4) and caused Ekman current towards the coast (Fig.
5c). Northerly or northeasterly winds caused opposite effects. Sea level was lower near the coast
compared to offshore and thermocline was located at shallower depths near the coast. Thus, the flow
was directed to the south in the surface layer. Such events occurred in late March and mid-August.
Most of the major events of the positive $V_{ADCP}$ and $V_{GETM}$ were associated with the positive $u$-
component of the Ekman current (cf. Fig. 5a and c), i.e., flow towards the shore, not along the shore.
Thus, the wind-driven strong coastal current to the north is not induced by the direct momentum flux
created by wind stress but rather is the result of wind-driven sea level gradient and depression of the
pycnoclines at the coast, which resulted in vertically sheared geostrophic current.
Next, we consider the relationship between the vertical maxima of the current shear and the vertical
location of pycnoclines – seasonal thermocline and halocline. Seasonal thermocline began to develop
from the beginning of May (Fig. 6a). The temporal course of salinity at 67 m depth (Fig. 6b) and depth
of halocline center (CH) (Fig. 6d) showed that halocline was mostly located deeper than the deepest
ADCP bin. At the end of March, the halocline center reached 55 m depth (Fig. 6d) and high current
shear values were observed below 45 m depth (Fig. 6c). Shallower halocline was related to the
northerly wind event (Fig. 4), which caused offshore Ekman transport in the upper layer and
compensating onshore flow in the deep layer (Fig. 3). Such events of high current shear in the deep
layer also occurred at the end of April to early May, from the end of May to mid-June and in mid-
August (Fig. 6c) when the halocline center was shallower, and salinity increased at 67 m depth. Note
that the depth of the halocline center and shear maxima were vertically shifted, halocline center was
deeper. This can be explained by the vertical range of the halocline. The upper boundary of the
halocline is shallower than the center of the halocline. Thus, the shear maxima were rather linked to
the upper boundary of the halocline.
Stronger and more extensive shear maxima in the upper part of the water column were observed since
late April (Fig. 6c). It appeared days before thermal stratification developed. One could see that SST
(sea surface temperature) and temperature at 67 m depth coincided until the end of April. The
occurrence of earlier shear maxima could be explained by the formation of the stratification in the





upper layer caused by the transport of fresher surface water to the area due to northerly wind forcing.
Shear maxima became stronger in the second half of May when thermal stratification developed.
Strong downwelling and likely also vertical mixing occurred in July as a result of a strong
southwesterly wind impulse with the duration of more than a week (Fig. 4). This can be seen as a drop
in SST from 21 to 15 °C and occasional high temperature recordings in the deep layer (Fig. 6a). The
latter indicates that the upper layer water arrived at the 67 m deep measurement spot. This event is well
reflected in the time series of current shear. Deepening of the shear maxima down to 50–55 m depth
(Fig. 6c) occurred together with thermocline deepening, as the near-bottom temperature recordings
suggest. Relaxation of the downwelling occurred in mid-July, and another downwelling developed at
the end of July. The linkage between the thermocline and shear maxima was well seen in August when
glider observations were available. The thermocline and shear maxima reached down to 40 m depth in
the beginning and the end of the month, while they were located at 20 m depth in the middle of the
month (Fig. 6a and c). The vertical movements of the halocline (Fig. 6d) and thermocline and linked
shear maxima were synchronized. As thermocline, the halocline had its position also shallower in mid-
August and deeper before and after. Note that downwelling was initiated by strong southerly,
southwesterly or westerly winds and all events were seen as a SST decrease, likely due to vertical
mixing, decrease in salinity at 67 m depth and deepening of the thermocline and halocline and related
shear maxima. Relaxation of downwelling occurred when northerly winds or calmer periods prevailed
and appeared as an increase in SST and upward movement of both pycnoclines.
Thus, we can conclude that the vertical structure of currents was strongly linked to the varying depths
of pycnoclines, which were sensitive to wind forcing.
**Table 1.** Statistics of the 1-h average ADCP current data from 28 February to 2 September 2020.

| Depth (m) | Mean speed (cm s$^{-1}$) | Mean $u$ (cm s$^{-1}$) | Mean $v$ (cm s$^{-1}$) | Maximum speed (cm s$^{-1}$) | Persistency (%) |
|---|---|---|---|---|---|
| 10.8 | 11.3 | 3.8 | 1.1 | 48 | 35.1 |
| 20.8 | 10.2 | 4 | 1.7 | 44 | 42.3 |
| 30.8 | 9.5 | 3.7 | 1.4 | 38 | 41.7 |
| 40.8 | 9 | 3.4 | 1.1 | 37 | 40.1 |
| 50.8 | 8.8 | 2.9 | 0.8 | 35 | 34.5 |
| 60.8 | 8.3 | 2.7 | 0.7 | 36 | 34 |
| 66.8 | 7 | 1.9 | 1.2 | 34 | 32.7 |









**Figure 3.** Temporal course of the low-pass filtered (36 h) current velocity *u*-component (positive eastward, a and c) and *v*-component (positive northward, b and d) in the water column (a, b); and in the upper (11 m depth) and deep layer (67 m depth, c, d) in the ADCP and Valeport locations in 2020 (Fig. 1). Low-pass filtered (10 days) wind 10°-component and current 40°-component at 10 m depth in the ADCP location (e).

**Figure 4.** Time series of the 10-m level ERA5 wind data from 1 March to 31 August 2020. Four selected periods
are shown: 1) prevailing southwesterly wind, 1–21 March; 2 and 3) prevailing northerly wind, 27 May–4 June
and 10–25 June; 4) prevailing southwesterly wind, 2 July–10 July. The green dotted line marks the beginning
and red dashed line marks the end of the period. Wind data were smoothed with a 36-h filter. Color scale shows
wind speed in m s$^{-1}$.






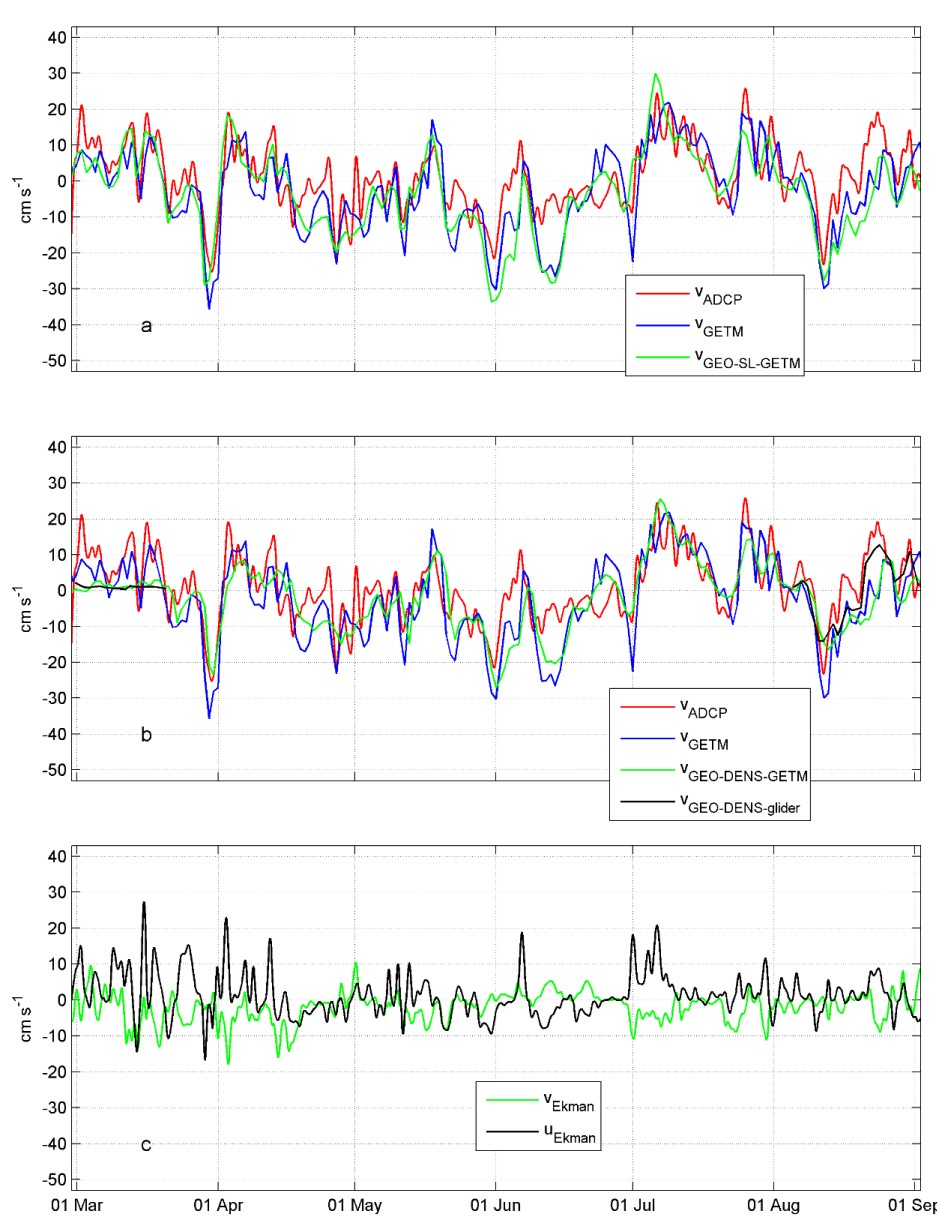

**Figure 5.** Temporal courses of (a, b panel) current velocity $v$-component measured by ADCP ($V_{ADCP}$), simulated
$v$-component ($V_{GETM}$), estimated from the GETM sea level data ($V_{GEO-SL-GETM}$), estimated from temperature and
salinity data collected by glider ($V_{GEO-DENS-glider}$), estimated from temperature and salinity data simulated by
GETM at 11 m depth ($V_{GEO-DENS-GETM}$). Mean Ekman current $u$-component and $v$-component ($U_{Ekman}$ and $V_{Ekman}$)
in the depth range 0–11 m (c). Time-series are shown from March to September 2020 at the ADCP location (see
Fig. 1).

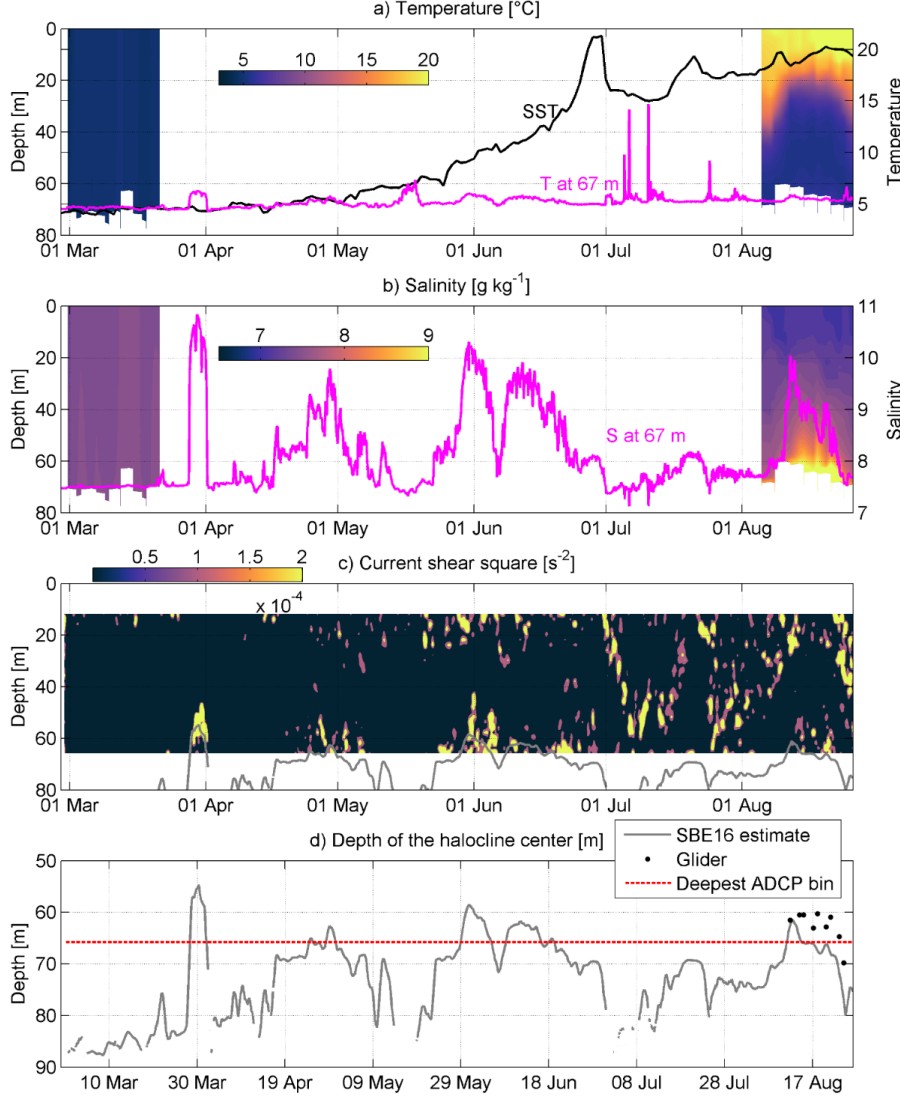

**Figure 6.** Temporal courses of temperature, salinity, current shear squared and halocline depth in the ADCP
location from March to September 2020 (see Fig. 1b and c). (a) Temporal course of sea surface temperature
(SST) and temperature at 67 m depth; temporal course of the vertical distribution of mean temperature in March
and August calculated from glider data. (b) Temporal course of salinity at 67 m depth; temporal course of the
vertical distribution of mean salinity in March and August calculated from glider data. Mean temperature and
salinity profiles were calculated for each glider passing within the 3.7 km zonal window around the ADCP
location. (c) Temporal course of the vertical distribution of current shear squared and depth of the halocline
center (grey line). (d) Depth of halocline center, calculated from SBE16 data and in August from glider data.
Depth of deepest ADCP bin is also shown (red dotted line).





## 3.2 Quasi-permanent circulation patterns

In the previous chapter, we demonstrated the importance of wind forcing and stratification for the currents. Next, we describe the current structure during the quasi-steady forcing periods. We have selected four periods of 8–21 days duration with relatively stable forcing (see Fig. 4) to analyze the mean measured and simulated flow structure in the ADCP and Valeport location (Fig. 7) and along the zonal section (Fig. 8). Likewise, we investigated the lateral simulated flow structures in the three forcing cases in three layers: upper layer (5 m), intermediate layer (40 m) and deep layer (110 m) (Figs. 9–11).

The persistency of the currents was very high in all selected periods (Table 2). Only during the fourth period, the persistency was lower than 50% below the seasonal thermocline. Particularly high persistency (82–94%) occurred in the first and second periods. Thus, currents during the quasi-steady forcing have much higher persistency than overall of the time series (see Table 1).

Barotropic flow to the northeast prevailed throughout the water column at the ADCP location in the first period (1–21 March) when south-westerly wind prevailed (Fig. 7a and b). Even stronger mean current to the north-northwest was registered at 5 m depth at the Valeport location (Fig. 3c and d). Latter indicates the boundary effect near the Saaremaa Island. The current was directed along the coast. Mean flow was to the south in the upper layer during the second period (27 May–4 June) when northerly wind prevailed to the southeast below the thermocline and to the east below the halocline (Fig. 7e and f). In general, a similar current pattern occurred in the third period (10–25 June) when north-westerly wind prevailed (Fig. 7i and j). Due to relatively strong south-westerly wind forcing in the fourth period (2–10 July), flow to the northeast prevailed in the upper layer and to westerly directions below the thermocline (Fig. 7m and n).

In conclusion, a pattern typical for the downwelling event – current to the northeast along the boundary and towards the shore in the upper layer and seaward current to the southeast in the deep layer – occurred during southwesterly wind domination (Fig. 7f and j). The flow was to the south in the upper layer along the coast and onshore (east) in the deep layer, which is typical for the upwelling cell in the case of northerly winds (Fig. 7n). These vertical patterns of the current velocity were also well captured by the numerical model (Fig. 7g, k and o), although the magnitude of the mean simulated velocity occasionally deviated from the measured values. Likewise, the stronger mean measured current near the boundary at the Valeport location, was well reproduced by the model (Fig. 7b and c). Geostrophic velocities had a quite similar vertical structure compared to the measured velocities in all periods (Fig. 7, third and fourth columns). Thus, currents were generally in geostrophic balance during the quasi-steady periods. The transition from one state to another has likely an ageostrophic nature, as wind is the main driver for the change.

Next, we analyze the vertical (Fig. 8) and horizontal (Fig. 9–11) structure of the mean meridional component of currents in the section along the latitude of the ADCP location (Fig. 1) and in the Eastern Gotland Basin using simulated current data. The current data are averaged within the same time windows with relatively stable wind forcing as analyzed above.

The structure of the meridional component of currents in the section is characterized by high spatial and temporal variability (Fig. 8). The unidirectional flow prevailed in most of the section down to the halocline or even deeper in the case of no thermal stratification and southwesterly winds (first period) (Fig. 8a). The northward current along the eastern boundary with a cross-coast extent of 10 km was especially strong. This strong boundary current was also registered by the Valeport (Fig. 3d). The strong maxima of the northward flow can be found between 20.5°–21.0° E, 18.6°– 19.3° E and around





17.6° E. The strong southward flow prevailed between 21.0°–21.3° E, 19.4°–20.0° E, and 17.6°–18.6°
E. Horizontal flow structure in the Eastern Gotland Basin consisted of the two stronger current zones
above the halocline, northward current along the eastern boundary and southward current in the middle
part (Fig. 9a and b). The two zones were connected with several cyclonic cells. The northward flow
below the halocline (Fig. 9c) coincided with the flow in the upper layer in the Eastern Gotland Basin
area but forced to the westward trajectory by bathymetry in the northern area.
The flow patterns were very similar in the following two periods (second and third) of prevailing
northerly winds and the presence of thermocline. In both cases, the zonal scale of the southward flow
around the ADCP location was 10–15 km (Fig. 8b and c). The flow did not extend to the eastern
boundary, a narrow northward flow with a width of 5–10 km occurred along the coastal slope. The
width of the southward flow near the western boundary of the section was about 30 km. In between,
several circulation cells with zonal scales of 20–60 km can be distinguished in the cross-section (Fig.
10a). The horizontal structure of the flow below the thermocline in the Eastern Gotland Basin revealed
a strong southward current in the eastern part of the area in the second period (Fig. 10b). The current
swirled, split into two branches and re-merged back to one in several locations. The southward flow
below the thermocline coincided with the offshore branch in the upper layer in the central area of the
basin (Fig. 10a and b). Sub-halocline flow revealed strongest northward current and strongest cyclonic
cell in the Eastern Gotland basin among the selected periods (Fig. 10c).
The flow pattern in the case of strong southwesterlies dominance (fourth period) under stratified
conditions revealed a strong northward current along both boundaries of the section (Fig. 8d). In
between, the strong southward flow occurred in the surface layer. Similarly, to the northerly wind
prevailing, complicated three-layer structure with variable horizontal patterns in the zonal scale of 20–
60 km occurred. Flow to the southeast prevailed in the upper layer, except in the eastern boundary
zone, where a strong northward downwelling related flow occurred (Fig. 11a), as also was observed in
our ADCP mooring data (Fig. 7n). A strong current occurred also in the Irbe Strait towards the Gulf of
Riga. Downwelling related flow along the eastern coast was also observed at 40 m depth (Fig. 11b). In
the deep layer below the halocline, northward current along the eastern bottom slope and cyclonic cells
in the Eastern Gotland Basin were observed (Fig. 11c).
Due to seasonality in forcing, variations in the circulation in this time scale can be expected. The
boundary current in the eastern coast occurs year-round but is the strongest in winter. This is related to
the wind regime: southwesterly winds prevail more in winter but are less frequent in spring and
summer. The seasonal signal can be found in the whole section (Fig. 12). Well defined large cyclonic
gyres in the Eastern Gotland Basin can be found in winter, while in spring and summer, the mean
current structure is characterized by the smaller zonal scale features and weaker flow. However, it is
noteworthy that the mean flow is to the north along the eastern coastal slope in all seasons.

**3.3 Sub-halocline current**
Cyclonic gyre was present below the halocline in the Eastern Gotland Basin in all selected periods
(Figs. 9–11). The flow in this cyclonic system was especially strong along the eastern slope of the
Eastern Gotland Basin. The northern branch of this circulation system is connected to the clearly
distinguishable northward current. The position and magnitude of the current varied under different
conditions. The current was stronger and meandered to west at the shallower area between Gotland and
Fårö Deep in the case of northerly wind while it was slower, and the meandering did not occur in the
case of southwesterly winds. To confirm the simulated cyclonic circulation in the Eastern Gotland





Basin and the northward flowing current towards the Northern Deep, the Argo float trajectory and the
mean current field were plotted in the same time frame (Fig. 13a). The general features in the simulated
mean currents and the Argo float trajectory agreed well. The Argo float first completed two circles
(smaller and larger) in the Eastern Gotland Basin and then headed to the north. The float arrived and
was recovered in the shallower area between the Fårö and Nothern Deep. This sill is an important
location for the deep layer water renewal in the Northern Baltic Proper (see bathymetry in Fig. 14), as
this is the only remarkable passage to the north below 100 m depth. The sill is located slightly south of
the selected section along the latitude of the ADCP deployment.
The flow to the north over the still was concentrated in a narrow cell with a zonal scale of 5–6 km (Fig.
15a). The flow was especially strong when northerly winds prevailed, e.g., in the second period from
27 May to 4 June (Fig. 15b). The 2010–2020 mean density field sloped downward in the left (west) of
the flow, typical for a gravity current (Fig. 15a–b). The meridional current velocity ($C_T$) in the trench
was mostly positive (northward) and in the range of 10–20 cm s$^{-1}$ during the study period in 2020 (Fig.
15c). The $C_T$ was reversed in the first half of July, which coincided with the strong southwesterly wind
impulse (Fig. 4). The time series of $C_T$ for 2010–2020 (Fig. 15d) revealed many reversal events, but
the long-term mean meridional velocity was 10 cm s$^{-1}$ to the north. Reversals were most frequent in
November–December when the monthly mean southward $C_T$ was 6–7 cm s$^{-1}$ and rarer in March–May
when monthly averages were in the range of 12–14 cm s$^{-1}$. Thus, the deep layer water renewal in the
Northern Baltic Proper is most active in the spring period and more restricted in late autumn–early
winter. The best correlation ($r^2$=0.25, $p<10^{-100}$, n=3838) between 10-day low-passed current velocity
at the sill and wind was found with the wind from ENE (70°) with a delay of 6 days. This is another
confirmation that prevailing southwesterly winds slow down or reverse the $C_T$ and prevent deep water
renewal in the Northern Baltic Proper.

**Table 2.** Persistency (%) of the currents at the selected depths during the selected periods: 1 March to 21
March (1); 27 May to 4 June (2); 10 June to 25 June (3); 2 July to 10 July (4) in 2020.

| Period/depth (m) | 1 | 2 | 3 | 4 |
|---|---|---|---|---|
| 10.8 | 84.8 | 82 | 75.8 | 83.1 |
| 20.8 | 88.8 | 92.3 | 76.9 | 78.9 |
| 30.8 | 88.8 | 94 | 66.2 | 54.8 |
| 40.8 | 88.6 | 92.5 | 62.1 | 41.3 |
| 50.8 | 89.3 | 89.9 | 61.4 | 24 |
| 60.8 | 87.7 | 91.1 | 70.1 | 27.5 |
| 66.8 | 87.2 | 86.1 | 64.1 | 4.7 |




**Figure 7.** The mean resultant wind vectors (a, e, i, m), mean profiles of current velocity vectors calculated from ADCP data (black arrows, b, f, j, n), mean current velocity vector based on Valeport data at 5 m depth (b, red arrow), mean simulated current velocity vectors at the ADCP location (c, g, k, o) and at the Valeport location (c, red arrow) are shown for selected periods (Fig. 4). On the right panels, mean adjusted geostrophic velocity vectors V$_{GEO-ADJ-GETM}$ (d, h, i, q) are shown.



**Figure 8.** Vertical distribution of mean meridional current velocities for four selected periods (see Fig. 4) along the ADCP deployment latitude (Fig. 1b). Color scale displays meridional velocity (positive northward) in cm s$^{-1}$. Vertical dotted lines show the ADCP location.




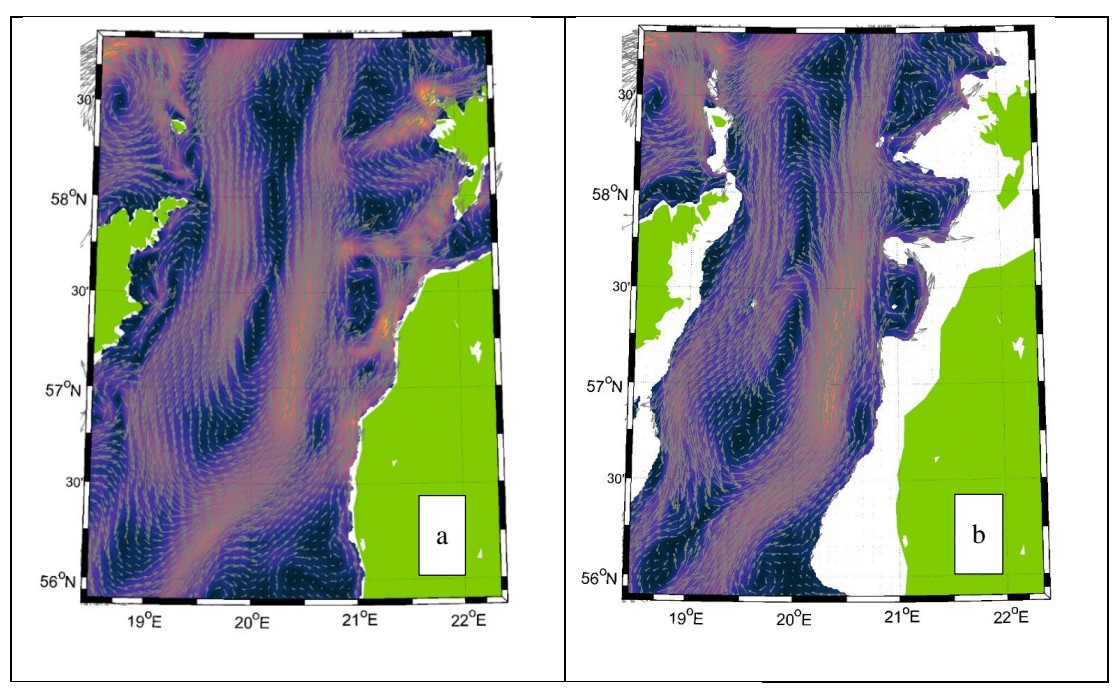

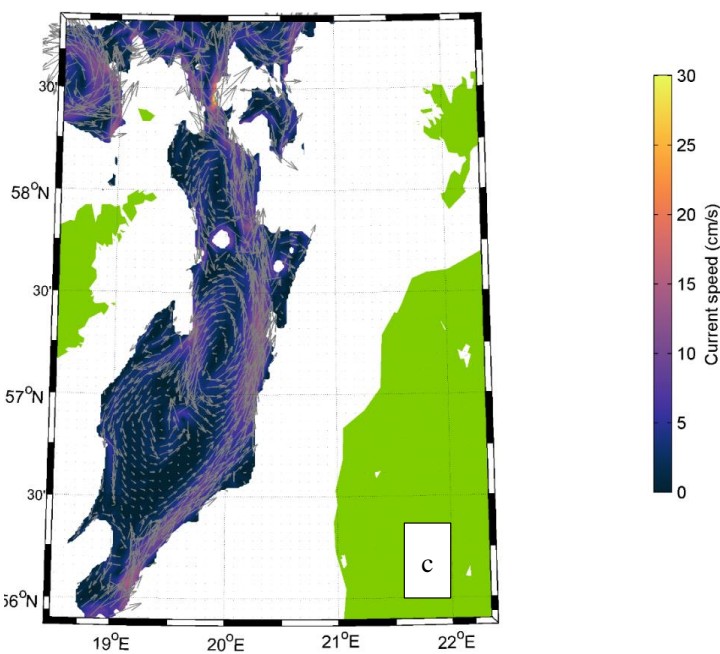

**Figure 9.** Mean currents in the case of prevailing south-westerly winds from 1 March to 21 March 2020, without thermocline at 5 m depth (a), 40 m depth (b) and 110 m depth (c). Color scale shows current speed in cm s$^{-1}$.







**Figure 10.** Mean currents in the case of prevailing northerly winds from 27 May to 4 June 2020, with
thermocline at 5 m depth (a), 40 m depth (b) and 110 m depth (c). Color scale shows current speed in cm s⁻¹.





**Figure 11.** Mean currents in the case of prevailing south-westerly winds from 2 July to 7 July 2020, with
thermocline at 5 m depth (a), 40 m depth (b) and 110 m depth (c). Color scale shows current speed in cm s$^{-1}$.





**Figure 12.** Vertical distribution of monthly mean (April, July and December) and annual mean meridional velocities (positive northward) along the zonal section at ADCP latitude based on simulation data from September 2010 to August 2020. Color scale shows meridional velocity in cm s$^{-1}$. Vertical dotted lines show the ADCP location.






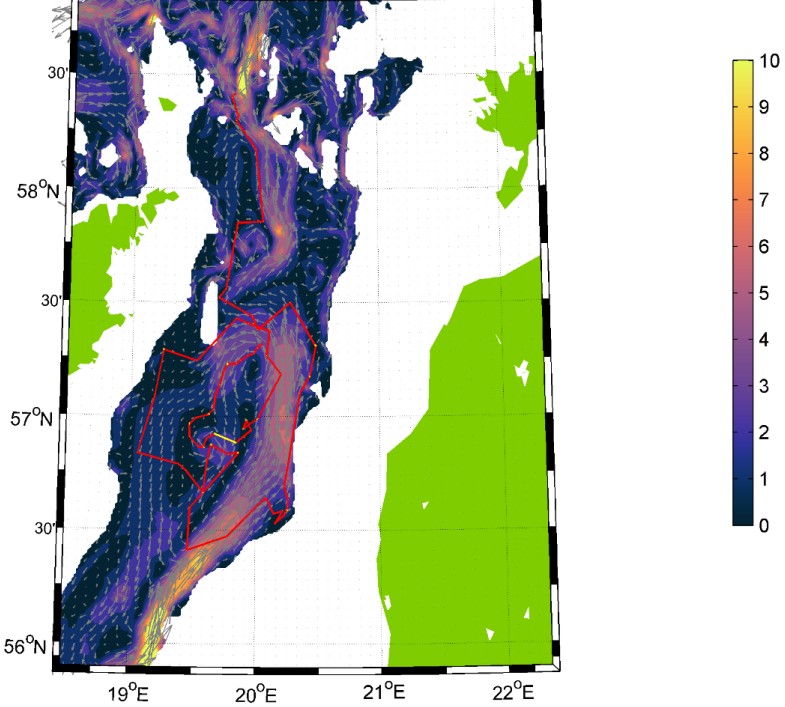

**Figure 13.** Mean current field between 105–135 m depth based on simulation data and ARGO float trajectory
during the period 15 August 2013–15 August 2014 in the deep layer (105–135 m, shown in red). Only one longer
period occurred, when the float drifted on the surface (shown in white). Color scale shows current speed in cm
s$^{-1}$.


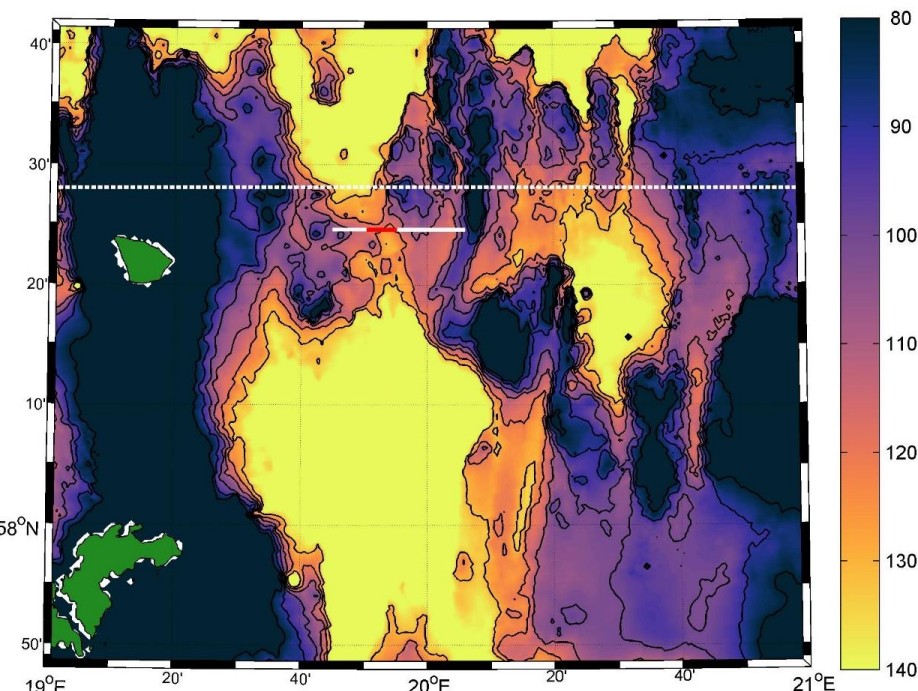

**Figure 14.** Bathymetry between Farö Deep and Northern Deep (see Fig. 1b). Color scale shows the depth in meters. White dashed line marks the section along the ADCP deployment latitude (Fig. 1b). White line marks the section in Fig. 15a, and red line indicates time-series calculation range for Fig. 15b–c.



**Figure 15.** (a) mean simulated meridional current component *v* and density isolines at section below 105 m depth (the section location is shown as red line in Fig. 14) in 2010–2020, (b) mean meridional current component *v* and density isolines at section below 105 m depth from 27 May to 4 June 2020 during a northerly wind impulse. In color scale contours with step of 2 cm s$^{-1}$ show current *v*-component (m s$^{-1}$, positive northward) and blue lines show density isolines with a step of 0.05 kg m$^{-3}$. (c) time-series of *v* component below 105 m at the sill. Dots marks the daily mean and bold line 10-day smoothed *v*-component from March to September. (d) time-series of *v* component below 105 m at the sill. Dots marks the daily mean, bold black line 10-day smoothed and bold blue line 3-month smoothed *v*-component in the period 2010–2020.





**4 Discussion**

Moorings carrying ADCP and single-point current meter, and underwater glider surveys were applied,
together with numerical modelling to investigate circulation in the Baltic Proper.
Strong linkage between the vertical location of the current shear maxima and the two pycnoclines was
observed. The same finding was reported in the Gulf of Finland (Suhhova et al., 2018). The current
shear maxima in the Gulf of Finland were related to the along-gulf estuarine circulation and its
alterations. In the present case, the shear maxima were related to the currents along the basin axis and
the coastal downwelling and upwelling circulation structures. The separation of the cross-shelf flow
by a pycnocline has been documented in several other coastal systems (Davis, 2010; Gilcoto et al.,
2017; Villacieros-Robineau et al., 2013).
Boundary current in the upper layer along the eastern coast was observed. The current was well
correlated with the wind. The wind regime in the area is the combination of the global circulation and
specific direction-dependent boundary-layer effects, which results in domination of winds along the
axis of the Baltic Proper (Soomere & Keevallik, 2001). Along-axis wind causes the Ekman current
(Ekman, 1905) to the right from wind direction in the upper layer, i.e., a flow across the basin axis.
The resulting convergence (divergence) in the case of southwesterly (northerly) winds at the eastern
coast causes across-axis sea level gradient and the upper pycnocline inclination, which in turn cause
horizontal pressure gradient, and results in a geostrophic flow to the north (south) in the upper layer.
Boundary currents forced by the pressure gradient caused by wind-driven divergence/convergence are
common in coastal systems (Berden et al., 2020; Longdill et al., 2008; H. Wu et al., 2013). The
geostrophic current velocity is well agreed with the total current velocity profiles. Thus, the current
along the boundary was generally in the geostrophic balance, but across-shore ageostrophic flow
created preconditions for this geostrophic coastal current.
Circulation rapidly reacted to the wind forcing. Persistency of the current for 6 months was rather low
(30–40%) due to variability in the wind forcing. The estimated persistency from long-term numerical
simulations data in the same area above the halocline was 70–80% in 1981–2004 (Meier, 2007) but
around 30–40% in the upper layer in 1958–2007 (Jędrasik & Kowalewski, 2019). However, the quasi-
steady circulation patterns detected under different wind and stratification conditions were high-
persistent, mostly >75%.
The mean cyclonic circulation in the upper layer of the Baltic Proper has been reported by many
modeling studies (Hinrichsen et al., 2018; Jedrasik et al., 2008; Jędrasik & Kowalewski, 2019; Meier,
2007; Placke et al., 2018). However, the magnitude of the long-term mean circulation patterns had a
considerably lower magnitude than the quasi-steady circulation structures presented in this study.
Likewise, the current direction of quasi-steady patterns varied and differed considerably from the long-
term mean. The circulation structures in this timescale also differ from the long-term mean because of
seasonal and inter-annual variations in the forcing. The cyclonic circulation and the eastern boundary
current towards the north in the upper layer is stronger in autumn and winter, as noted by previous
simulations (Jędrasik & Kowalewski, 2019), when strong southwesterly winds are more frequent
(Soomere & Keevallik, 2001). Quasi-steady circulation patterns were characterized by complicated
lateral vortices with the zonal scale of 20–60 km. The richness of vortical structures has been suggested
by several numerical modelling studies (Dargahi, 2019; Zhurbas et al., 2021). In-situ measurements





are needed to verify the existence of the vortices and to characterize their effect on the physical and
biogeochemical fields in more detail.
Two quasi-permanent circulation features were detected in the deep layer. Cyclonic gyre was present
below the halocline in the Eastern Gotland Basin, with the strongest flow along the eastern slope, which
has been documented by in-situ measurements earlier (Hagen & Feistel, 2004; Hagen & Feistel, 2007).
The northern branch of the Eastern Gotland Basin current is connected to the quasi-steady northward-
flowing current towards narrow Fårö sill between the Fårö and Nothern Deep. The width of the current
was mostly 10–30 km, but only 5 km at the sill. The mean northward component of the current was 10
cm s$^{-1}$, which can be explained by the mean density structure (Fig. 15a) and is typical for the gravity
current in a channel (Zhurbas et al., 2012). This current is an important deeper limb of the Baltic haline
conveyor belt (Döös et al., 2004). The current was stronger in the case of northerly winds and weaker
during southwesterly wind prevailing. This is typical behavior of the estuarine circulation: up-estuary
wind causes weakening or reversal of the deep layer current and down-estuary wind intensification of
the estuarine current (Geyer & MacCready, 2014) as observed in the Gulf of Finland (Liblik et al.,
2013; Lilover et al., 2017; Suhhova et al., 2018) and several other estuaries (e.g. Giddings &
MacCready, 2017; Scully, 2016). In the case of northerly wind, the vertical and horizontal density
gradient in the Fårö sill was much stronger (Fig. 15b) than the mean gradient in 2010–2020 (Fig. 15a)
according to the simulation. Note that on the right-hand flank, the isopycnals are vertical (Fig. 15b). A
similar structure of the gravity current has been measured by acoustic profiling in the Western Baltic
(Umlauf et al., 2009). The current to the north and potentially the deep layer water renewal in the
Northern Baltic Proper is more intense in March–May when southwesterly winds are less frequent, and
the current is weakest in November–December. If the water that overflows the Fårö sill is dense
enough, it occupies the Northern Deep bottom layers, and the old, oxygen-depleted bottom water is
lifted and advected to the Gulf of Finland, as observed during high Major Baltic Inflow activity (Liblik
et al., 2018). If the overflow has a lower density compared to the deep layer waters in the Northern
Deep, it does not dive to the bottom but stays as a buoyant layer.

The most favorable wind for the up-estuary deep layer advection in the Gulf of Finland is from the
northeast (Elken et al., 2003). Thus, northerly winds support deep water renewal and strengthening of
the stratification all the way from the Gotland Deep to the Gulf of Finland. The deep layer currents are
quite well covered by observations in the Gulf of Finland (Lilover et al., 2017; Rasmus et al., 2015;
Suhhova et al., 2018). However, observations are lacking from the Gotland Deep to the entrance of the
Gulf of Finland. The only in-situ record about the feature between Gotland and Northern Deep is the
Argo float track. The Argo trajectory supported our suggestion about the existence of the sub-halocline
current to the north. Our simulations suggested that the strength and position of the current did depend
on the wind forcing. Observations and simulation results at the channel-like topographic constriction,
Slupsk Furrow, in the southern Baltic have shown that the meandering of the gravity current is strongly
affected by the bottom topography and wind-forcing (Zhurbas et al., 2012). ADCP measurements are
needed to understand the behavior of the sub-halocline current better.
Overall, simulated currents quite well agree with the ADCP measurements in the upper layer. However,
the meridional component of the simulated current ($V_{GETM}$) was biased (Fig. 5a). The mean $V_{ADCP}$ was
1.1 cm s$^{-1}$, but the mean $V_{GETM}$ was –3.2 cm s$^{-1}$ at 10 m depth during the study period. Such bias could
not be found in the deep layer. Flow to the north was often weaker compared to measurements ($V_{ADCP}$),
and flow to the south was stronger than observed by the ADCP in the upper layer. A similar tendency
can be found in a comparison of the ADCP measurements and simulation results in the Gulf of Finland





(Suhhova et al., 2015). Near the right-hand side coast (looking up-estuary, i.e., to the east in the Gulf
of Finland), the down-estuary flow was stronger and more frequent in the simulation compared to the
measurements (see their Fig. 2). Interestingly, a similar bias was detected in the deep layer at the eastern
flank of the Gotland Deep at 204 m depth (Placke et al., 2018). Four different models considerably
underestimated (Placke et al., 2018) the mean flow to the north derived from observations (Hagen &
Feistel, 2004). The first possible explanation for the bias could be the smaller width of the boundary
current. Indeed, the mean flow towards north in 2010–2020 was stronger in the east from the ADCP
location (Fig. 12). The second possible source for the discrepancy could be related to the performance
of simulation of ageostrophic or geostrophic flow. We will discuss this further in the next section.
Quite large discrepancies between the simulation and the measurements occurred in June. In the first
half of the month, simulation was biased to the south, but in the second half, a bias to the north can be
seen (Fig. 5a). In both cases, the geostrophic current seems to play an important role in the discrepancy.
Strong simulated $V_{\text{GEO-DENS-GETM}}$ to the south (north) occurred in the first (second) part of June. In
August, the simulation did not capture the strongest flow event to the north on 21–24 August (Fig. 5a).
At the same period, much lower values of the $V_{\text{GEO-DENS-GETM}}$ compared to the $V_{\text{GEO-DENS-glider}}$ can be
seen. These signs suggest, first, that the isopycnals in the model react to the forcing more rapidly than
in the sea. Secondly, there is a bias in the across/slope seasonal thermocline inclination. Likely, the
thermocline is tilted more towards the surface near the coast in the model than in the sea. We next
evaluate the measured (by glider) and simulated temperature, salinity and geostrophic velocity fields
on 11–12 August and on 22–23 August.
Surface layer geostrophic velocity in the simulation agrees well with the estimates from the glider data
on 11–12 August (Fig. 16a–b). Though, the glider observations reveal sharper thermocline inclination
than the simulation. Discrepancies in the temperature, density, and geostrophic current fields on 22–
23 August are much larger (Fig. 16c–d). Glider observations revealed the thermocline depressed down
near the coast, which is typical for a downwelling. The inclination in the thermocline caused strong
geostrophic flow to the north in the location of ADCP (Fig. 16c). Homogenous mixed layer reached
down to 22 m depth at the easternmost end of the section. Such an inclination, well defined
homogenous layer and geostrophic current to the north at the ADCP location was not revealed by the
simulation (Fig. 16c). Thus, we can conclude that the bias in the boundary current simulation could be
related to the inaccuracy of reproducing the temperature and salinity fields and the resulting
geostrophic component of currents. We are not going into further details of this problem here, as it is
out of the focus of the present work. However, conclusions of the simulation studies that have focused
on the long-term mean current fields in the upper layer, but did not validate simulations with direct
current observations, should be taken carefully, as the magnitude of the long-term residual current is
very small compared to the magnitude of the currents during the quasi-steady states. We suggest a
dedicated study involving numerous current profiling records should be conducted to track down the
causes of the discrepancies between observations and simulations.


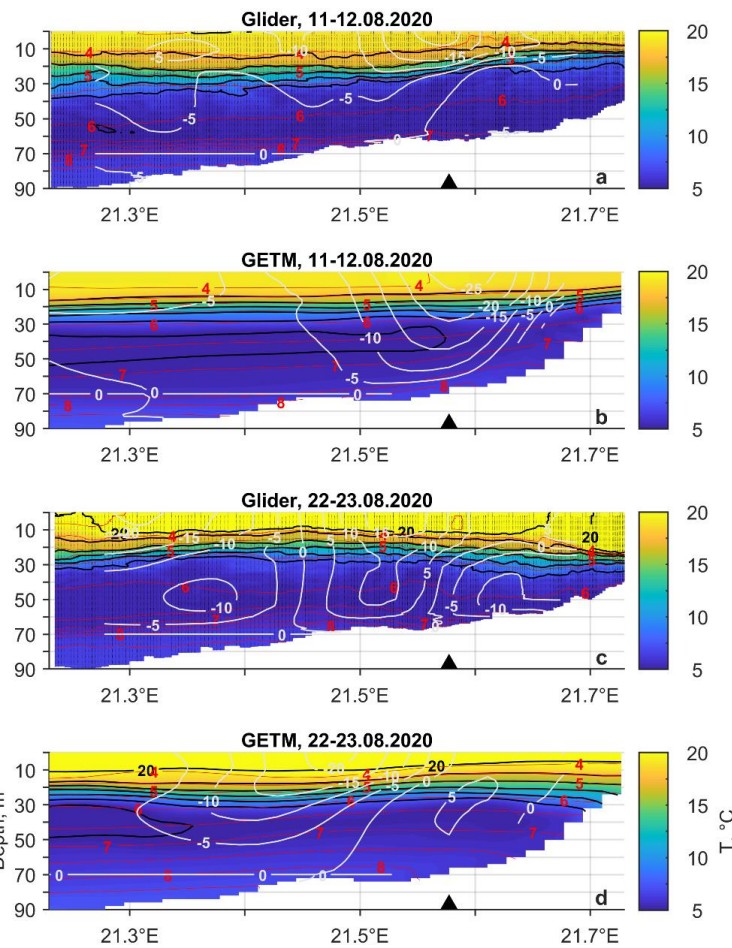

**Figure 16.** Temperature (color contours), density isolines (red lines), relative geostrophic current (white lines) based on glider observations and GETM simulation on 11–12 August and 22–23 August 2020.

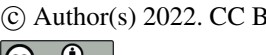



## 5 Conclusions

A strong link between the existence and location of the two pycnoclines and the current structure was observed. Boundary current was observed in the upper layer along the eastern coast of the Baltic Proper. The current was mainly in geostrophic balance, but across-shore Ekman transport created preconditions for the geostrophic coastal current. The boundary current rapidly reacted to the changes in the wind forcing that is reflected in a relatively low persistency of currents (30–40%) in the whole water column during the 6-month measurement period. However, the quasi-steady circulation patterns formed under the certain wind and stratification conditions were high-persistent (mostly >80%) and generally in the geostrophic balance.

The sub-halocline, quasi-steady northward (towards Fårö sill) gravity current with a width of 10–30 km was detected by the simulation. The finding was supported by the Argo float displacement data. This important deeper limb of the Baltic Sea haline conveyor belt is stronger in the case of northerly winds and weaker during south-westerlies. More detailed studies of the dynamics and water properties of this current are essential to understand the renewal process of deep layer waters in the Northern Baltic Proper and in the Gulf of Finland.

Generally, the structure of boundary current was well reproduced by the GETM. However, the meridional component of the simulated current was biased southward. Further investigations of the current regimes in various locations during the periods of quasi-steady forcing could help to reveal the causes of the discrepancy.

*Code availability*. Scripts to analyze the results are available upon request. Please contact Taavi Liblik.

*Autor contributions*. TL led the analyses of the data and writing of the paper with contributions from GV, JL, UL, KS and MJL. TL was responsible for the measurements and data processing, and GV for the modelling activities. KS processed the glider data.

*Competing interests*. The authors declare that they have no conflict of interests.

*Acknowledgements*. We would like to thank our colleagues and research vessels Salme crew for all the support in measurements and operations at sea. The computing time from high-performance computing center at Tallinn University of Technology and University of Tartu are gratefully acknowledged. GETM community at Leibniz Institute of Baltic Sea Reasearch are gratefully acknowledged for maintaining and developing the code.

*Financial support*. This work was supported by the Estonian Research Council grant PRG602. Collection of the data was financially supported by the European Regional Development Fund within National Programme for Addressing Socio-Economic Challenges through R&D (RITA). Infrastructure assets used in the current study are part of the JERICO infrastructure and supported by the JERICO-S3 project under the European Union's Horizon 2020 research and innovation programme with grant number 871153.





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
