# Peer review of "Quasi-steady circulation regimes in the Baltic Sea"

_Ocean Science, 2021_

## Author Response (AR1)

**Reviewer 1**

In this paper the data from several measurement methods (adcp- and current profiler mooring, on board CTD, glider missions and argo floats) are applied together with model simulations to further study the circulation patterns in the Baltic Proper. The topic is of interest and merits further research, as especially the currents on the deeper layers are not yet too well known.

Response: We thank for your time and constructive review!

The dataset and analysis provided here is a valid addition that increases our understanding of the conditions in the area and in my opinion well merits the publication with some minor additions. Data and methods of analysis are reasonable and the conclusions drawn from them valid. In general the text and presentation is clear. Few minor issues could use further work:

The combination of modelled results and measurements is an interesting part of the work. When considering the effects of bathymetry and model resolution, it would help to see the model grid size demonstrated with Figure 1 or (b or c perhaps) with grid overlay, or with 'measuring stick' to get a quick idea of the scale of details the model can catch.

Response: Yes, that kind of visualisation help reader to understand the visualisation.

Action: The model grid size is quite small for this large area, the grid in the sea area would hide the bathymetry information. Thus, we decided to add grid points to land. It gives idea about the model resolution but keeps bathymetry visible.

In line 134 the authors mention a qc method for removing suspicious/failed profiles. How many were there? Giving a percentage of accepted profiles would clear up the reliability of the device.

Response: Actually, we did not detect failed profiles during these missions.

Action: We added an explanation to the parenthesis „(impossible date and location test, range tests for the sensors; practically no incorrect data were detected)„.

In results, in the chapter starting from 289 authors compare the ADCP results with the ones given by the model, noting a southward bias in the model results. The agreement with the model and results is actually rather good, but I wonder could these biases be due to either resolution or bathymetry setup in the model. It might be worthwhile in discussion to ponder if tuning mode setup based on these findings could improve it further.

Response: That is a good question. We have dedicated three sections in discussion for that topic. Starting from „Overall, simulated currents quite well agree with the ADCP measurements in the upper layer." and finished in „We suggest a dedicated study involving numerous current profiling records should be conducted to track down the causes of the discrepancies between observations and simulations." Our conclusion is that the bias likely could be related to the inaccuracy of reproducing the temperature and salinity fields and the resulting geostrophic component of currents. We are afraid we cannot go deeper in the topic in the present paper. As we wrote in the final referred sentence, experiment with numerous current profiling stations will allow much more comprehensive handling of this issue. We

plan to conduct international current profiling experiment (several ADCP stations across Baltic Proper) in 2022. This data should help to clarify the issue.

Action: We rephrased the last sentence of conclusions; „Further *in-situ* measurements and simulations of the current regimes in various locations during the periods of quasi-steady forcing could help to reveal the causes of the discrepancy.“

Around Line 341 the authors discuss the movement of the thermocline depth. How was the thermocline depth determined in this case? It would be clarifying to see a similar description than in the case of halocline earlier.

Response: Actually thermocline was not quantitatively estimated. It is good idea to do it similarly to the halocline.

Action: We estimated thermocline depth similarly to the halocline. We inspected temperature profiles and defined isotherm where the temperature gradient was strongest (13 °C) as the center of the thermocline depth. We wrote this procedure to the methods chapter. We could only estimate the thermocline depth for the glider mission in August as thermocline did not exist in winter. Thermocline depth in August is visible in Figure 6a and 6c and these panels are now cited in text in respective places.

Few more comments which are more on the clarity of the manuscript, and perhaps a matter of taste:

-It might ease up the reading to state clearly when speaking of model data, when measurements, for example Fig 8-11, or in the chapter starting from 470. It is often clear from context, but for a quick reader it would help.

Response: Yes, that will clarify the source of the data for a reader

Action: We added „simulation“ to figure captures. We also modified sentence: „Next, we analyze the vertical distribution of monthly mean (April, July and December) and annual mean meridional velocity component along the zonal section at ADCP latitude based on simulation data from September 2010 to August 2020.“ there. We added „simulated“ and „measured“ to other appropriate locations in the text.

-Figure 1 text says "Study area (black box)" should probably be blue box.

Action:We fixed.

-In Figure 7 (line 503) there are so many sub-plots that the area for each gets a bit small. I wonder, could it be possible to join foir example ADCP + GETM + GEO-ADJ-GETM subplots together with different colors?

Response: We tried to make it as you suggested, but the result was not better.

Action: No action here.

-Figure 11 (or other which is near to fig 12) could mark the location of transect of Figure 12 for comparison.

Response: That is good idea.

Action: We added the line to a) panel in figures 9-11.

**Reviewer 2**

**The review of the manuscript 'Quasi-steady circulation regimes in the Baltic Sea' by Liblik et al. submitted for the publication in the Ocean Science.**

The submitted manuscript describes the ocean circulation patterns observed in the Baltic Proper in 2020 and the forcing mechanisms behind them. The authors use extensive set of new observational data from moored instruments (two current meters plus one CTD recorder), 2 glider missions, one Argo float, several CTD profiles and back them with the numerical model results and atmospheric reanalysis data.

**General comments**

The major concern in this kind of study is whether the data collected in a restricted area can be representative for a larger region, such is the Baltic Proper in this case. By using the model results and a long Argo trajectory the authors convinced me that the link between the point measurements, as well as rather short glider sections and more general circulation pattern does exist here. The obtained time series are not only thoroughly processed, quality controlled and analyzed but also deliberately matched with the model outcome. Several topics/processes are analyzed. The in situ data-model output comparisons are supported by additional observations and everything is illustrated by the appreciable number of figures (15) and two tables. The complex manuscript is well written and allows the reader to familiarize well with the specifics of the area and science problems. The received results are convincingly yet cautiously discussed in the light of the previous findings and the broader/long-term perspective. The conclusions drawn by authors are interesting and motivate to further studies. Thus, I have only a few questions and some comments that can potentially improve the way of the presentation. Otherwise, I have no more concerns and I suggest a minor revision.

Response: Thank you for your time and the constructive review! We agree the major concern is important question and there is room for improvement for next studies. Surprisingly, there is very limited current observation data available from the baltic Proper. That was one of the motivators of the current study. To our knowledge the present study is backed by one of the most comprehensive current observation dataset in the Baltic Proper: six months of ADCP measurements, 2.5 months of point current meter measurements, two months of glider measurements and one year Argo drift. Many previous circulation studies did not use circulation measurements at all, but relied solely on simulations.

Action: We modified the last sentence of conclusion to better highlight the importance observations in circulation studies. "Further *in-situ* measurements and simulations of the

current regimes in various locations during the periods of quasi-steady forcing could help to reveal the causes of the discrepancy."

**Specific comments**

Page 2, line 52: for the pelagic ecosystem - if it concerns the deep bottom layer it probably also impacts the benthic ecosystem. Consider adding this.

Response: Yes, we agree.

Action: We removed pelagic. It reads now „The conveyor determines salinity, stratification and other important characteristics for the ecosystem."

Page 2, line 57: increase hypoxia in the Northern Baltic Proper and Gulf of Finland - why? Please add 'due to ....' that the reader does not need to look for this information elsewhere.

Response: Yes, we can do it.

Action: It reads now „Only Major Baltic Inflows (Matthäus & Franck, 1992; Mohrholz, 2018) ventilate the deep layers of the southern and central Baltic Proper (Holtermann et al., 2017) but increase hypoxia in the Northern Baltic Proper and Gulf of Finland due to the transport of former anoxic/hypoxic Eastern Gotland Basin water and creating stronger stratification (Liblik et al., 2018)."

Page 8, line 236: Persistency of the current - what does the persistence tell us? Consider adding a few words, like 'informs about… and is defined by'

Response: It can be added.

Action: It reads now: „Persistency of the current, characterizing the variability of the direction of the flow, is defined as the ratio between vector and scalar current speeds:"

Page 9, line 270: The flow resulting from the sea level gradient and due to the inclination of isopycnal surfaces are also a consequence of wind but develop slower - Nicely explained!

Response:Thank you!

Action: No change.

Page 9, line 276: 0.6 m/s - this is rather low wind speed, do not you think? Is this a mean for all months from this period? Or for Mar-Aug only? The impact of seasonality could be mentioned here, I think.

Response: This is not mean wind speed, but mean wind component towards 10-degrees. Please see the explanation in the previous section. It is good idea to mention seasonality.

Action: We added „The $w_{10}$ is higher in winter and smaller in summer. Considering the linear relation between the two variables, the 1979–2020 mean $w_{10}$ = 1.1 m s$^{-1}$ corresponds to $c_{40}$ = 4.2 cm s$^{-1}$.“

Page 11, line 331: a drop in SST from 21 to 15 °C – this is interesting and a bit counterintuitive. In other parts of the Baltic Sea such a fierce drop in surface temperature in summer is often a sign of upwelling, not the downwelling. What is the source of this cold water – the mentioned vertical mixing and cooling alone? I would expect that northerlies are able to cool the sea surface more efficiently than the southwesterlies unless they are much stronger. A set of SST maps from late June/early July would make it clear (possible advection path), I think.

Response: Yes, it is vertical mixing mostly behind the event. Also, some cooling probably occurred. We have checked the SST maps. It happened in a large area, not only in the study area. Important here is that 21 °C is exceptionally high SST for this region at the end of June. There was a very strong atmospheric heat flux to the sea before the mixing event. The weather was sunny and the air temperature was high. Also, the wind was very weak, see figure 4. This allowed to form a thin and warm surface layer, which however was easily mixed with colder subsurface water during the strong wind impulse event. We don't want include heat flux etc. calculations in the manuscript as that is not the focus of the paper, but we add explanation for such a rapid drop.

Action: We added sentence: „A precondition for such a rapid drop in SST was the formation of a thin and exceptionally warm surface layer due to atmospheric heat flux (Fig. 6a) and weak wind (Fig. 4) at the end of June.“

Page 16, line 416: occasionally deviated from the measured values - no surprise, it would be strange for the model to show the same results as in observations all the time.

Response: We agree, it is not surprising

Action: We removed that part of the sentence.

Figures

Figure 4 - Are you still able to change the color palette? I think this one is not color-blind friendly.

Response: Thank you for noting.

Action: We changed the color palette to color/blind friendly.

Figure 5 - Similarly here, it would be nice to avoid red and green color together.

Response: We tried to make it better.

Action: We changed red to black and made blue lighter.

Figure 8 and 12 – Could you add the notation about geographical sites: W and E or Sweden and Estonia? It would help to grasp the bathymetry/orientation at once.

Response: Yes, that will make it easier for reader to follow.

Action: We added W and E.

Figure 13 – What is the parking depth for this float? Slightly above the bottom or the same along the whole Argo trajectory (~100 m)? What information does the ANDRO product provide here? The same as Argo GDAC? Could you, please provide WMO for this float?

Response: It was between 105-135 m. ANDRO provides the displacement data (when and where it surfaced and parked). It was convenient to derive trajectory from the database. We did not download the data from the Argo GDAC.

Action: We added the parking depth and WMO number to the figure caption.

**Technical corrections**

Page 2, line 48: so called - this is an informal phrase, use another one.

Response: We agree.

Action: We removed it.

Page 9, line 278: at the Valeport location - this can be a sentence start, stressing the change in instrument being described.

Response: Yes, that would make it easier for reader to follow.

Action: It reads now „At the Valeport location, the most frequent current direction was 350°.“

Page 10, line 288: low-passed filtered - low-pass filtered

Response: We agree.

Action: We fixed in the whole manuscript.

Page 10, line 289: reasonably well – put it at the end of the sentence

Action: We did so.

Page 10, line 299: evoked – induced

Action: We changed.

Page 16, line 405: northerly wind prevailed – should not there be a coma here?

Response: We think you are right.

Action: Added coma.

Page 16, line 412: The flow was to the south in the upper – rephrase a bit to make this sentence similar to the previous one, like 'On the contrary, a pattern typical for the upwelling ....' and continue in similar way as before. This would make things easier for the reader, because there are many directions and layers in this description and it is easy to get lost.

Response: We agree it makes it easier to follow.

Action: We changed as suggested.

Page 16, line 414: These vertical patterns - do you mean downwelling and upwelling? If yes, say it (e.g. in brackets).

Response: We agree

Action: We changed as suggested.

Page 16, line 422: Next, we analyze the vertical (Fig. 8) and horizontal (Fig. 9–11) structure - this part is somehow disconnected from the previous one. You need to clarify why do you include it. Say something like: 'To understand (what?)....we next analyze the vertical...'. Similarly, you can explain why do you want to analyze model data in this area (the Eastern Gotland Basin) that is out of the area where all of the measurements were taken (this is the major concern I mentioned in my general comments).

Response: Yes, we can introduce this section a bit better. The reason for analyzing the model data out of the measurement area is to understand the larger-scale circulation dynamics better. We showed that during the quasi-steady periods current structure is quite well reproduced by simulation. We believe this allows to provide trustworthy picture of the larger scale current dynamics as well. The conclusion about subhalocline current was supported by the Argo float trajectory.

Action: It reads now „Next, to understand the larger scale circulation dynamics during the periods, we analyze the vertical".

Page 17, line 437: but forced – but was forced?

Action: We fixed.